# Diagnostic accuracy of triglyceride to glucose index and triglyceride/high-density lipoprotein index for insulin resistance among children and adolescents: A systematic review

Miguel Cabanillas-Lazo[1◉], Carlos Quispe-Vicuña[1◉], Milagros Pascual-Guevara[1‡], Claudia Cruzalegui-Bazán[2,3‡], Arturo Duran-Pecho[2,3‡], José Paz-Ibarra[3,4‡], Victor Velásquez-Rimachi[1*]

1 Grupo de Investigación Neurociencias, Metabolismo, Efectividad Clínica y Sanitaria (NEMECS), Universidad Científica del Sur, Lima, Perú, 2 Sociedad Científica de San Fernando, Lima, Perú, 3 Facultad de Medicina de San Fernando, Universidad Nacional Mayor de San Marcos, Lima, Perú, 4 División de Endocrinología, Hospital Nacional Edgardo Rebagliati, Lima, Perú

◉ These authors contributed equally to this work.
‡ MP-G, CC-B, AD-P and JP-I also contributed equally to this work.
* vvelasquezr@cientifica.edu.pe

## Abstract

### Introduction

Insulin resistance (IR) is a common metabolic disorder associated with obesity, type 2 diabetes, and cardiovascular diseases. There is a need for simpler and more cost-effective indices to diagnose IR in pediatric populations. While the hyperinsulinemic-euglycemic clamp is the gold standard for diagnosing IR, it is costly and complex. Consequently, simpler indices like the triglyceride-to-glucose (TyG) and triglyceride/high-density lipoprotein (Tg/HDL) indices have been explored as alternative diagnostic tools for IR.

### Methods

This systematic review adhered to PRISMA-DTA guidelines. Analytical studies evaluating the diagnostic accuracy of TyG and Tg/HDL indices for IR in children and adolescents were included. Eligibility criteria required studies to report sensitivity, specificity, or area under the curve (AUC) values compared against reference standards for IR. Excluded studies were those without reported diagnostic accuracy measures. Data were extracted from PubMed, Scopus, Web of Science, Embase, and SciELO Citation Index up to April 2024. Due to high heterogeneity, a narrative synthesis was performed.

**Data availability statement:** All relevant data related to the search strategy, study selection, and evidence evaluation are provided in the Supplementary Material to ensure reproducibility. No additional data are available.

**Funding:** Author receiving the award: VVR (Victor Velásquez-Rimachi) Grant number: N/A Funder: Universidad Científica del Sur Funder's website: https://www.cientifica.edu.pe The funders had no role in the study design, data collection and analysis, decision to publish, or preparation of the manuscript.

**Competing interests:** The authors have declared that no competing interests exist.

## Results

Twenty-one studies involving 28,768 participants were included. The TyG index showed sensitivities ranging from 60–92% and specificities of 54–100%, with AUCs between 0.610 and 0.960. For the Tg/HDL index, sensitivities varied from 14.8–85.7%, specificities from 60.9–97.6%, and AUCs from 0.687 to 0.809. These results suggest that the TyG index generally demonstrates higher diagnostic precision than the Tg/HDL index, but both exhibit moderate accuracy with considerable heterogeneity.

## Conclusion

The TyG and Tg/HDL indices are promising tools for diagnosing IR in children and adolescents, but further studies with rigorous methodology are needed. Future research should focus on standardizing cut-off points and exploring their predictive value in diverse populations.

## Introduction

Insulin resistance (IR) is a metabolic condition caused by a reduction in the ability of cells to absorb and utilize blood glucose [1]. IR results in altered insulin signaling and glucose entry into adipocytes and skeletal muscle cells. While the cause of IR is not fully understood, pathophysiological mechanisms contributing to this are oxidative stress, insulin receptor mutation, endoplasmic reticulum stress and mitochondrial dysfunction [1]. The development of IR seems to be closely related to excess body weight, and the incidence of IR is parallel to obesity and the incidence of type 2 diabetes. Although precise global data are lacking, IR is approximately much more common than diabetes, with an estimated prevalence of approximately one quarter of the world's population, equivalent to more than one billion people [2]. Additionally, it has been associated with the development of multiple diseases, such as metabolic syndrome [3], type 2 diabetes mellitus [4], and cardiovascular diseases [5], among others. IR can affect people of all ages. However, due to the increase in overweight and obesity in the young population in recent decades [6], it has become a public health problem among children and adolescents [7].

This underlines the importance of having an effective diagnostic technique to establish adequate treatment. The hyperinsulinemic euglycemic clamp (HEC) test is currently the gold standard for the diagnosis of IR in the pediatric population. This test determines tissue sensitivity to insulin (hepatic and muscular) and β-cell response to glucose. However, the HEC test is expensive and complex to perform, and thus, indirect methods, such as the homeostasis model assessment index for insulin resistance (HOMA-IR) or fasting plasma insulin are frequently used [7,8], because they are easier to perform. However, for their calculation insulin and glucose levels are determined in fasting plasma, but due to the instability of insulin in the patient, the blood collected must be processed and frozen immediately [9]. All of this involves

a great deal of logistics on the part of the hospital, which often cannot be adequately achieved, especially when a large number of samples are required [10]. Moreover, fasting insulin measurement is not a routine test in obese children, the HOMA-IR cut-off point is not fully established, and HOMA-IR has not been correlated with IR [7,8]. HOMA2 and QUICKI are alternative markers to HOMA-IR for assessing insulin resistance. HOMA2 accounts for nonlinear glucose-insulin dynamics, while QUICKI shows a strong correlation with the HEC test [1]. These indices can provide dynamic assessments but require multiple samples, making them less practical due to standardization and accessibility challenges.

All this leads to the need for an effective diagnostic test that is easier to perform and inexpensive. In this context, triglyceride (Tg), high density lipoprotein (HDL) and glucose levels are of interest and are routinely determined using inexpensive tests compared to those evaluating IR; indeed portable analyzers are now available to calculate these values in a simple manner [11]. For these reasons, the use of the Tg/HDL index and Tg-Glucose (TyG) index as cost-effective predictors or markers of IR in children and adolescents has been reported in different regions [9,12,13].

Therefore, we developed this systematic review to evaluate the evidence regarding the diagnostic accuracy of the Tg/HDL and TyG indices in the diagnosis of IR in children and adolescents.

## Methods

This systematic review was reported according to the Preferred Reporting Items for Systematic Reviews and Meta-Analyses of Diagnostic Test Accuracy Studies (PRISMA-DTA statement) (S1 Table) [14]. The study protocol was registered in PROSPERO with the code CRD42021287129.

### Eligibility criteria

Studies were included if they met all the following criteria: 1) Analytical observational studies (cross-sectional and cohort studies); 2) Studies including children or adolescents (aged ≤18 years) with or without obesity, and 3) Studies that evaluated the diagnostic accuracy of the TyG index and Tg/HDL index or at least one of them. There were no restrictions on language or publication dates.

The following formulas for TyG were considered:

$$\text{Formula A}: \ln[\text{Tg (mg/dL)} \times \text{fasting glucose (mg/dL)}/2]$$

$$\text{Formula B}: \ln[\text{Tg (mg/dL)} \times \text{fasting glucose (mg/dL)}]/2$$

The following formula for Tg/HDL was considered:

$$\text{Formula C}: \text{Tg(mg/dL)}/\text{HDL (mg/d)}$$

### Data sources

We searched in PubMed, Embase, Scopus, Web of Science and the SciELO Citation Index until April 2024. The search strategy for PubMed was adapted for use in the other databases (S2 Table). We completed the search by reviewing the bibliographic references of the studies included and selecting the articles that met the requirements.

### Study selection

The electronic search results were imported into the Endnote X9, and duplicate records were excluded. Then, these records were exported to the Rayyan (https://rayyan.qcri.org/). A peer review process was performed by two reviewers

(MPG and CCB) and any discrepancies were resolved by consensus and with the opinion of a third reviewer (MCL). These reviewers assessed inclusion criteria independently by reading the full texts of the potentially relevant studies selected and discrepancies were resolved by consensus.

## Data collection process

Two authors (MPG and CCB) independently carried out data extraction using a data extraction form, and any disagreements were resolved by consensus and ultimately a third author (MCL). We extracted the following information: title of the study, first author, year of publication, study design, country where the study was performed, number of participants, sex, age, values of sensitivity and specificity, index and reference test cut-off, area under curve (AUC), positive predictive value (PPV) and negative predictive value (NPV). If the PPV or NPV were lacking, they were calculated using the following formulas: PPV was determined by dividing the number of true positives by the sum of true positives and false positives, while NPV was calculated by dividing the number of true negatives by the sum of true negatives and false negatives. Disagreements were resolved by consensus.

## Risk of bias and applicability

The quality of the studies was assessed with the Revised Tool for the Quality Assessment of Diagnostic Accuracy Studies (QUADAS-2) [15] by two authors (MCL and CQV). This tool evaluates four domains: (1) selection of patients; (2) conduction and interpretation of the index test; (3) type and interpretation of the reference standard (considered optimal when it consisted of the HEC); and (4) patient flow.

Bias in patient selection was addressed by excluding studies with non-representative populations. For the index test, we ensured that TyG and Tg/HDL indices were measured consistently with clear definitions. Reference standard bias was minimized by including studies that used validated methods such as HOMA-IR or HEC. Finally, flow and timing issues were mitigated by including only studies with consistent protocols for index and reference testing. Any disagreements in bias assessment were resolved by consensus.

## Narrative synthesis

Due to the high methodological and clinical heterogeneity, no meta-analysis was performed; for this reason, we applied a narrative synthesis. Narrative synthesis is an approach of multiple studies, including systematic reviews, that use words and text to summarize and explain the findings of the synthesis. We used a four-stage process based on guidelines: [16] (1) Describe the pathophysiology or biological-clinical plausibility of the TyG and the Tg/HDL indices to diagnosis IR; (2) Developing a preliminary description (sensitivity, specificity, AUC and participants) of the index test in the study populations; (3) Summarizing the diagnostic accuracy data in tables according to types of index test and population; (4) Evaluating the certainty of evidence using the Grading of Recommendation, Assessment, Development, and Evaluation (GRADE).

## Certainty of evidence assessment

Two authors (CCB and CQV) assessed the certainty of evidence with the GRADE approach by narrative synthesis [17]. This assessment is based on six domains: study design, study limitations (risk of bias), imprecision [sample size and confidence interval [CI]), indirectness (generalizability), inconsistency (heterogeneity), and publication bias as stated in the GRADE handbook [18]. There are some reports of adaptation assessing the certainty of evidence of narrative synthesis in diagnostic tests but without systematic, clear or transparent decision making [19–22]. We adapted the assessment to our results according to the index test (TyG and Tg/HDL) and population (obese/overweight and general population); for this reason, for our systematic review we used a self-developed decision table (S3 Table). The certainty of the evidence was characterized as high, moderate, low, or very low.

## Results

### Study selection

We identified 2177 studies through our systematic search. We removed duplicates and screened 1135 studies and finally included 21 studies [9,13,23–41] (Fig 1). The list of all the studies evaluated in the selection process is provided in S4 Table. The complete list of excluded articles is provided in S5 Table.

### Characteristics of studies

Twenty of the studies included were cross-sectional and one was a cohort study. The total number of participants was 28,768. The reference tests used across the studies were the HEC, HOMA-IR index, and the Hyperinsulinemia and Matsuda index. (Table 1) shows the studies included.

The reference tests used across the studies were HEC, the gold standard for measuring insulin sensitivity, HOMA-IR, which estimates insulin resistance from fasting glucose and insulin levels, and the Matsuda index, which assesses whole-body insulin sensitivity using oral glucose tolerance test data. Definitions and cut-off values of IR were heterogeneous among the studies (Table 2).

### Risk of bias and applicability

According to the results of the QUADAS-2 tool, there was an overall moderate to high risk of bias. The patient selection domain had a low risk in almost all the studies, as was the index test domain. The flow and time domain presented low risk in all studies. The reference standard domain was somewhat at risk in almost all the studies, since most of the studies did not use a standard test for the diagnosis of IR. Applicability concerns were low in the patient selection, index test, and reference standard domains of all the studies. Fig 2 presents a graphical summary of the overall risk of bias assessment of the studies included in this review. In addition, S6 Table describes the risk of bias results for each QUADAS-2 domain for each study.

### Narrative synthesis

The diagnostic accuracy of both the TyG index and the Tg/HDL index has been extensively studied in participants who are obese or overweight. When evaluating the TyG index, the studies used HOMA-IR as a reference, employing two different formulas for calculation (Table 3).

Studies using Formula A reported cutoff values ranging from 7.9 to 8.66 with sensitivities of between 60% and 92%, and specificities of 54% to 100%, in a population of 777 participants. Another study utilizing the Matsuda index as a reference found a sensitivity of 85% and a specificity of 61%, with an AUC of 0.75 [13]. Studies using Formula B reported cutoff values of 4.2 to 7.9, with sensitivities of between 65% and 89%, specificities of 58% to 100%, PPV of 54.9% to 64.9%, and NPV of 64.2% to 87.7% in a population of 772 participants. However, there was notable heterogeneity in the AUCs reported, ranging from 0.610 to 0.960. In the general population, the diagnostic accuracy of the TyG index has been assessed using the same two formulas. Studies using Formula B reported TyG index cutoff values of 4.7 to 8.5, with sensitivities between 65% and 88.5% and specificities of 74.3% to 78.1%, based on a larger cohort of 2980 participants [34]. Conversely, studies using Formula A presented cutoff values ranging from 7.9 to 8.3, with sensitivities between 62% and 84% and specificities from 50.5% to 81%, drawn from a substantial sample size of 6052 participants. The AUCs for these studies varied from 0.640 to 0.860.

In relation to the Tg/HDL index, its diagnostic accuracy in obese or overweight participants has been explored across multiple studies, predominantly utilizing HOMA-IR as a reference (Table 4). Seven out of nine studies reported Tg/HDL index cutoff values of 1.36 to 3.0, with sensitivities between 14.8% and 85.7%, specificities from 60.9% to 97.6%, PPV from 45.6% to 87.9%, and NPV from 35.8% to 84.2% in 2721 participants. Notably, there was considerable heterogeneity in the AUCs reported, ranging from 0.687 to 0.809. Additionally, one study utilized HEC as a reference, revealing a cutoff

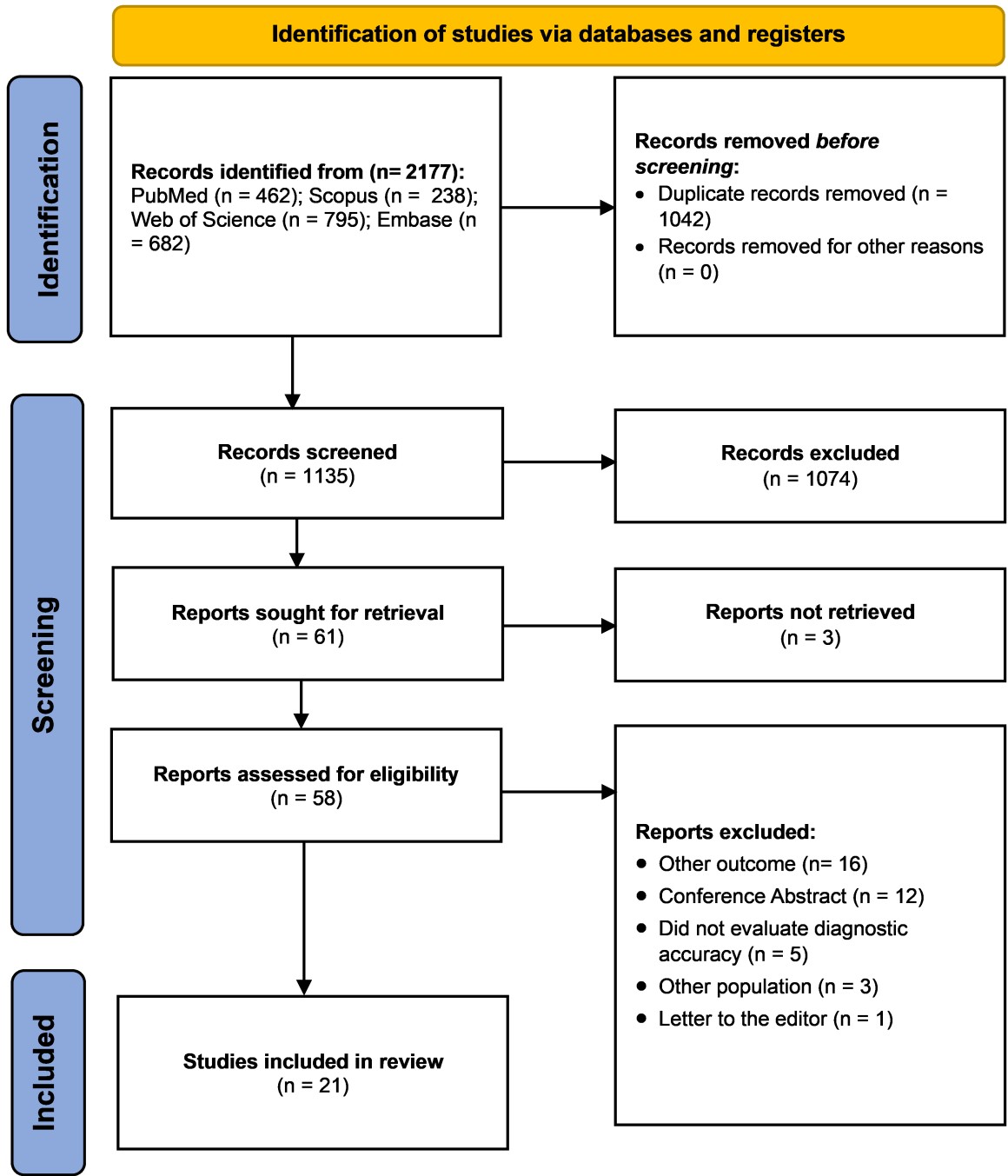

**Fig 1. PRISMA flowchart of the studies included.**

of 3 with a sensitivity of 61% and a specificity of 82%, and an AUC of 0.747 [31]. In the general population, the diagnostic accuracy of the Tg/HDL index has been evaluated in various studies, all using HOMA-IR as a reference. These studies reported Tg/HDL index cutoff values of 1.41 to 2.22, with sensitivities between 68% and 94%, specificities of 48% to 86%, PPV of 5.5% to 54.2%, and NPV of 13.7% to 95.3% based on 718 participants. The AUCs ranged from 0.729 to 0.81.

**Table 1. Characteristics of the studies included according to the reference standard of insulin resistance [n=21].**

| Study | Country | Study design | Population | Reference tests | Age [SD] | TyG [n] | Tg/HDL [n] | Reference [n] | Funding |
|---|---|---|---|---|---|---|---|---|---|
| *Obese or overweight population/patients* | | | | | | | | | |
| Hannon, 2006 [31] | USA | Cross sectional | Overweight adolescents | HEC | 13.50 [1.60] | NR | 35 | 35 | National Center for Research Resources |
| Giannini, 2011 [30] | USA | Cohort | Obese children and adolescents | WBISI | 13.10 [2.90] | NR | 1452 | 1452 | National Institutes of Health |
| Bridges, 2016 [25] | USA | Cross sectional | Obese and over-weight children and adolescents | HOMA-IR Hyperinsulin-emia | 13.40 [1.26] | NR | 223 | 223 | National Institutes of Health |
| Yoo, 2017 [32] | Korea | Cross sectional | Overweight and normal weight children and adolescents | HOMA-IR | O: 8.70 [2.00] N: 8.90 [1.80] | NR | 769 | 769 | NR |
| Calcaterra, 2019 [27] | Italy | Cross sectional | Children and adoles-cents of a Polyclinic for auxological evalua-tion of obesity | HOMA-IR | 11.70 [2.71] | 541 | NR | 541 | None |
| Behiry, 2019 [24] | Egypt | Cross sectional | Obese and overweight children | HOMA-IR | 10.10 [2.99] | NR | 90 | 90 | None |
| Locateli, 2019 [33] | Brazil Colombia | Cross sectional | Overweight and obese children and adolescents | HOMA-IR | 15.40 [1.80] | 345 | 345 | 345 | None |
| Dikaiakou, 2020 [13] | Greece | Cross sectional | Obese children | HOMA-IR Matsuda index | 9.90 [2.30] | 367 | NR | 367 | NR |
| Rodriguez, 2020 [34] | Mexico | Cross sectional | Overweight and obese children and adolescents attending consultory | HOMA-IR | 2-16 years* | NR | 628 | 628 | Clínica de Obesidad del Hospital Materno Infantil |
| Sanchez, 2020 [36] | Spain | Cross sectional | Obese children and adolescents | HOMA-IR | 12.10 [2.14] | 60 | NR | 60 | Universidad Alfonso X El Sabio |
| Yoon, 2022 [41] | Korea | Cross sectional | Obese and over-weight children and adolescents | HOMA-IR | 11.34 [3.24] | 176 | NR | 176 | NR |
| You-Xiang, 2023 [39] | China | Cross sectional | Obese children and adolescents | HOMA-IR | 14 [12.0-15.0]** | NR | 404 | 404 | Guangdong philosophy and Social Science Foundation |
| Zhang, 2024 [28] | China | Cross sectional | Obese children | HOMA-IR | 9.39 [1.82] | NR | 262 | 262 | Beijing Hospital Management Centre Paediatric |
| *General population/patients* | | | | | | | | | |
| Kang, 2017 [9] | Korea | Cross sectional | Children and adoles-cents at schools | HOMA-IR | 11.10 [1.50] | 221 | 221 | 221 | Ministry of Science of South Korea |
| Rodriguez, 2017 [35] | Mexico | Cross sectional | Children and adoles-cents at schools | HOMA-IR | 12.90 [2.40] | 2779 | NR | 2779 | Fundacion IMSS of Mexico |
| Alvim, 2018 [23] | Brazil | Cross sectional | Children and adoles-cents of schools | HOMA-IR | 10.70 [2.00] | 296 | 296 | 296 | Federal University of Espírito Santo |
| García, 2019 [29] | Mexico | Cross sectional | Children | HOMA-IR | 8.0 [5.0-9.0]** | 201 | 201 | 201 | Fundacion IMSS of Mexico |
| Brito, 2020 [26] | Brazil | Cross sectional | Children at schools or participating in a support program | HOMA-IR | 7.80 [1.28] | 515 | NR | 515 | National Council for Scientific and Techno-logical Development |

*(Continued)*

**Table 1.** (Continued)

| Study | Country | Study design | Population | Reference tests | Age [SD] | TyG [n] | Tg/HDL [n] | Reference [n] | Funding |
|---|---|---|---|---|---|---|---|---|---|
| Song, 2021 [37] | Korea | Cross sectional | Children and adolescents | HOMA-IR | 14.60 [0.06] | 3728 | NR | 3728 | None |
| Hirschler, 2022 [38] | Argentina | Cross sectional | Children at school | HOMA-IR | 9.27 [2.17] | 915 | NR | 915 | Cariño Study Group |
| Reckziegel, 2023 [40] | Brazil | Cross sectional | Children and adolescents at schools | HOMA-IR | 12.79 [1.96] | 377 | NR | 377 | NR |

*Interval range; ** Median and interval.

TyG: Triglyceride-glucose index; Tg/HDL: Triglyceride/high-density lipoprotein cholesterol index; HEC: Hyperinsulinemic euglycemic clamp; NR: Not reported; HOMA-IR: Insulin resistance index; O: Overweight; N: Normal weight; WBISI: Whole body insulin sensitivity index.

**Table 2. Summary of the reference tests [n=21].**

| Study | Reference test | Insulin resistance definition | Insulin resistance cutoff value |
|---|---|---|---|
| Hannon, 2006 [31] | HEC | 80 mU/m2/min | NR |
| Giannini, 2011 [30] | WBISI | <10th percentile | NR |
| Bridges, 2016 [25] | HOMA-IR Hyperinsulinemia | Top quartile >25u IU/mL | NR |
| Kang, 2017 [9] | HOMA-IR | ≥95th percentile | NR |
| Yoo, 2017 [32] | HOMA-IR | ≥3 | ≥3 |
| Rodriguez, 2017 [35] | HOMA-IR | ≥ 95th percentile | NR |
| Alvim, 2018 [23] | HOMA-IR | ≥3.16 | 3.16 |
| Behiry, 2019 [24] | HOMA-IR | ≥4.0 | NR |
| Calcaterra, 2019 [27] | HOMA-IR | > 97.5th percentile | NR |
| Locateli, 2019 [33] | HOMA-IR | NR | ≥3.16 |
| Brito, 2020 [26] | HOMA-IR | >75th percentile | NR |
| Dikaiakou, 2020 [13] | HOMA-IR Matsuda index | ≥3 <br> ≤2.5 | ≥3 <br> ≤2.5 |
| García, 2020 [29] | HOMA-IR | ≥90th percentile | 8.23 |
| Rodriguez, 2020 [34] | HOMA-IR | ≥2.5 | ≥2.5 |
| Sanchez, 2020 [36] | HOMA-IR | Previous literature | Previous literature |
| Song, 2021 [37] | HOMA-IR | ≥ 95th percentile | NR |
| Hirschler, 2022 [38] | HOMA-IR | > 3rd quartile | NR |
| Yoong, 2022 [41] | HOMA-IR | NR | NR |
| You-Xiang, 2023 [39] | HOMA-IR | >3 | NR |
| Reckziegel, 2023 [40] | HOMA-IR | ≥3.16 | 3.16 |
| Zhang, 2024 [28] | HOMA-IR | ≥4.0 | NR |

HOMA-IR: Insulin resistance index; NR: Not reported; HEC: Hyperinsulinemic euglycemic clamp.

## Certainty of evidence

With respect to the diagnostic efficacy of the TyG and Tg/HDL indices, we found the certainty of the tests included to be very low. We assumed a high certainty because almost all the studies were analytically observational. For the TyG index, the certainty was low by presenting a high risk of bias, with high inconsistency among the studies due to the presentation of very heterogeneous values for sensitivity (60% to 92% for obese/overweight patients and 62% to 88.5% for the general population) and specificity (54% to 100% for obese/overweight patients and 50.5% to

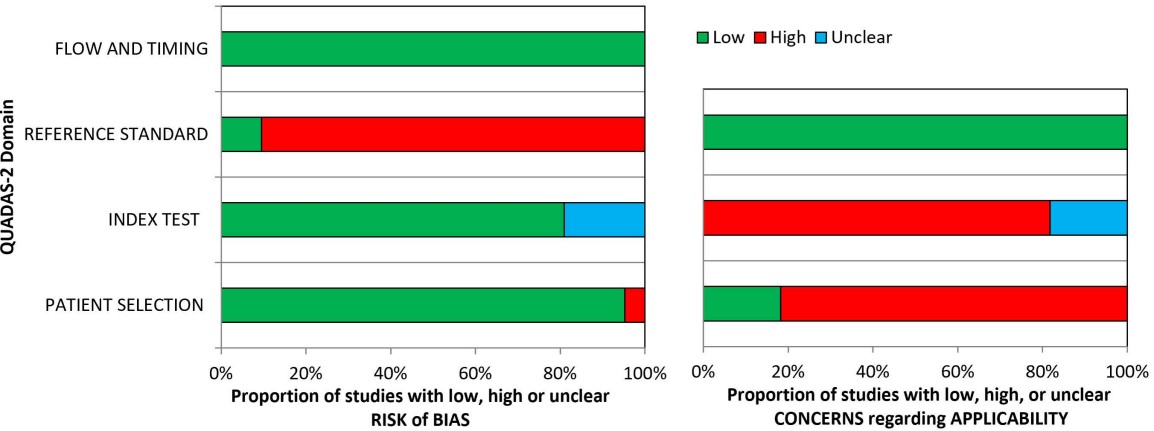

**Fig 2. Graphical summary of the risk of bias of studies included using the QUADAS-2 tool.**

81% for the general population). Regarding the diagnostic efficacy of the Tg/HDL index, certainty was low due to a high risk of bias, high inconsistency among studies, with very heterogeneous values for sensitivity (14.8% to 89% for obese/overweight patients and 45.0% to 97.6% for the general population) and specificity (45% to 97.6% for obese/overweight patients and 48% to 86% for the general population) (Table 5). The diverse reference standards used across studies create challenges in directly comparing diagnostic accuracy. This heterogeneity likely reduces the reliability of pooled estimates for sensitivity and specificity, underscoring the need for standardization in future research.

## Discussion

### Summary of the main results

Our study aimed to evaluate the diagnostic accuracy of the TyG and Tg/HDL indices in both obese/overweight participants and the participants from the general population, utilizing HOMA-IR as the reference test. This systematic review with meta-analysis (1527 participants included) revealed high heterogeneity among the studies assessing the TyG and Tg/HDL indices in both overweight/obese and general children and adolescent populations. In overweight/obese participants, the TyG index showed varying values of sensitivity (60%−92%) and specificity (54%−100%) with AUCs from 0.610 to 0.960, while the Tg/HDL index had a sensitivity of 14.8%−85.7% and a specificity of 60.9%−97.6% with AUCs from 0.687 to 0.809. For the general population, the TyG index had sensitivity of 62%−88.5% and a specificity of 50.5%−81% with AUCs from 0.640 to 0.860, and the Tg/HDL index had a sensitivity of 68%−94% and a specificity of 48%−86% with AUCs from 0.729 to 0.81. Despite their potential, the certainty of evidence for the diagnostic accuracy of these indices remains low.

### Methodological contribution

The GRADE system provides an option for evaluating the certainty of evidence for systematic reviews of intervention, diagnosis and prognosis [42–44]. However, these options are mostly based on meta-analyses. Although attempts have been made to evaluate narrative results without meta-analysis [45], these only establish points for evaluations of interventions. In the area of diagnostic accuracy, the proposals reported are not clear and with explicit parameters [19,20]. For this reason, we have created a decision table for the GRADE domains that can be taken or adopted for other reviews without meta-analysis.

**Table 3. Summary of the diagnostic accuracy measures reported for the triglyceride-glucose [TyG] index [n = 13].**

| Study | Reference test | Definition TyG | TyG cutoff | Sensitivity | Specificity | PPV | NPV | AUC |
|---|---|---|---|---|---|---|---|---|
| *Obese or overweight population/patients* | | | | | | | | |
| Calcaterra, 2019 [27] | HOMA-IR | ln[TG[mg/dL] × fasting glucose [mg/dL]/2] | 7.9 | 60.0 | 78.0 | NR | NR | 0.690 |
| Dikaiakou, 2020 [13] | HOMA-IR Matsuda index | ln[TG[mg/dL] × fasting glucose [mg/dL]]/2 | 7.9 | 65.0 85.0 | 58.0 61.0 | 58.8 54.9 | 64.2 87.7 | 0.650 0.750 |
| Locateli, 2019 [33] | HOMA-IR | ln[TG[mg/dL] × fasting glucose [mg/dL]/2] | 4.44 | 75.7 | 67.4 | 64.6 | 77.8 | 0.743 |
| Sanchez, 2020 [36] | HOMA-IR | ln[TG(mg/dL) × fasting glucose (mg/dL)/2] ln[TG(mg/dL) × fasting glucose (mg/dL)]/2 | Prepubertal: 7.9 Adolescents: 8.2 Prepubertal: 4.2 Adolescents: 4.3 | 92.0 72.0 84.0 89.0 | 100.0 54.0 100.0 69.0 | NR NR NR NR | NR NR NR NR | 0.960 0.700 0.840 0.610 |
| Yoong, 2022 [41] | HOMA-IR | ln[TG[mg/dL] × fasting glucose [mg/dL]/2 | 8.38-8.66 | NR | NR | NR | NR | 0.839 |
| *General population/patients* | | | | | | | | |
| Brito, 2020 [26] | HOMA-IR | ln[TG[mg/dL] × fasting glucose [mg/dL]/2] | 7.9 | 80.0 | 55.3 | 37.7 | 89.1 | 0.717 |
| García, 2020 [29] | HOMA-IR | ln[TG[mg/dL] × fasting glucose [mg/dL]]/2 | 8.5 | 65.0 | 74.3 | 21.7 | 4.9 | 0.802 |
| Kang, 2017 [9] | HOMA-IR | ln[TG[mg/dL] × fasting glucose [mg/dL]/2] | 8.18 | 77.4 | 64.8 | 19.5 | 96.3 | 0.734 |
| Alvim, 2018 [23] | HOMA-IR | ln[TG[mg/dL] × fasting glucose [mg/dL]/2] | NR | M:84 F: 83 | M:81 F: 63 | M:4.3 F: 11.7 | M:50 F: 47.6 | M:0.860 F: 0.760 |
| Rodriguez, 2017 [35] | HOMA-IR | ln[TG[mg/dL] × fasting glucose [mg/dL]]/2 | Prepubertal 4.7 Pubertal F: 4.8 Pubertal M: 4.7 | 83.1 88.5 88.0 | 74.5 78.1 76.8 | 43.0 36.9 36.3 | 86.9 92.5 92.2 | 0.721 0.747 0.745 |
| Song, 2021 [37] | HOMA-IR | ln[TG[mg/dL] × fasting glucose [mg/dL]/2] | 8.3 | 66.5 | 65.6 | 20.7 | 93.6 | 0.723 |
| Hirschler, 2022 [38] | HOMA-IR | ln[TG[mg/dL] × fasting glucose [mg/dL]/2] | 8.0 | 62.0 | 62.0 | NR | NR | 0.650 |
| Reckziegel, 2023 [40] | HOMA-IR | ln[TG[mg/dL] × fasting glucose [mg/dL]/2] | 7.94 | 75.0 | 50.5 | NR | NR | 0.640 |

PPV: Positive predictive value; NPV: Negative predictive value; AUC: Area under curve; HOMA-IR: Insulin resistance index; NR: Not reported; M: Male; F: Female; TG: triglycerides.

## Comparison with other systematic reviews

Our study showed that, despite very low certainty of evidence, Tg/HDL could be a good marker for determining insulin resistance. Taking into account that this condition generates metabolic changes such as dyslipidemia, inflammatory patterns, endothelial dysfunction, hypertension, and obesity [46], it has been reported that for each unit of glucose that increases, the risk of developing cardiovascular disease is 0.83 (95% CI 0.78–0.90, I2: 58.9%) [47], so it is important to know the role of Tg/HDL in this association. Due to the simplicity of the method, by dividing triglycerides by high-density lipoproteins only (Tg/HDL-C) and converting them into logarithms, a plasma atherogenic index is obtained, which has been reported as a novel biomarker of cardiovascular risk [48].

The potential utility of our studied biomarker is corroborated by Chen et al. [49] whose results revealed that individuals with the highest TG/HDL ratio had a significantly higher risk of cardiovascular events (combined HR: 1.43, 95% CI: 1.26–1.62, I² = 72.9%). This finding was consistent when analyzing the TG/HDL ratio as a continuous variable (HR grouped by unit increase in the TG/HDL-C ratio: 1.08, 95% CI: 1.04–1.12, I² = 67.0%)

**Table 4. Summary of the diagnostic accuracy measures reported for the Tg/HDL index [n=12].**

| Study – ID | Reference test | Tg/HDL cutoff | Sensitivity | Specificity | PPV | NPV | AUC |
|---|---|---|---|---|---|---|---|
| *Obese or overweight population/patients* | | | | | | | |
| Yoo, 2017 [32] | HOMA-IR | ≥2* | 55.6 | 72.9 | 45.6 | 80.0 | NR |
| Behiry, 2019 [24] | HOMA-IR | ≥1.36 | 85.7 | 66.7 | 69.2 | 84.2 | 0.809 |
| Bridges, 2016 [25] | HOMA-IR Hyperinsulinemia | 2.27 | 14.8 16.4 | 97.6 97.6 | NR 71.4 | NR 76.1 | 0.720 0.710 |
| Locateli, 2019 [33] | HOMA-IR | >2 | NR | NR | NR | NR | 0.687 |
| Rodriguez, 2020 [34] | HOMA-IR | 2.27 | 70.5 | 63.1 | 87.9 | 35.8 | 0.720 |
| Giannini, 2011 [30] | WBISI | 2.27 | 89.0 | 45.0 | NR | NR | NR |
| Hannon, 2006 [31] | HEC | ≥3 | 61.0 | 82.0 | NR | NR | 0.747 |
| You-Xiang, 2023 [39] | HOMA-IR | NR | 71.0 | 71.4 | 67.0 | 77.5 | 0.784 |
| Zhang,2024 (6–9 years) [28] | HOMA-IR | NR | 79.1 | 60.9 | 66.9 | 74.4 | 0.734 |
| Zhang,2024 (10–13,5 years) [28] | HOMA-IR | NR | 79.4 | 62.9 | 68.1 | 75.3 | 0.724 |
| *General population/patients* | | | | | | | |
| Kang, 2017 [9] | HOMA-IR | 1.41 | 72.7 | 61.8 | 17.4 | 95.3 | 0.736 |
| Alvim, 2018 [23] | HOMA-IR | NR | M: 68 F: 94 | M: 86 F: 48 | M: 7.8 F: 5.5 | M: 48.5 F: 52.8 | M: 0.81 F: 0.74 |
| García, 2020 [29] | HOMA-IR | 2.22 | 90.0 | 48.6 | 54.2 | 13.7 | 0.729 |

PPV: Positive predictive value; NPV: Negative predictive value; AUC: Area under curve; HOMA-IR: Insulin resistance index; NR: Not reported; WBISI: Whole body insulin sensitivity index; HEC: Hyperinsulinemic euglycemic clamp; M: Male; F: Female.

*Cutoff only for overweight patients.

The review by Baneu *et al.* [50], adhered to the PRISMA guidelines and assessed the potential of the Tg/HDL ratio as an indirect indicator of IR. Analyzing 32 studies conducted over 20 years, involving 49,782 participants from various ethnic backgrounds, including both adults and children, this review predominantly employed cross-sectional analysis using the HOMA-IR to measure IR. It revealed a variable predictive power of the Tg/HDL ratio across different ethnic groups and genders, with specific thresholds providing greater accuracy for Caucasians, Asians, and Hispanics compared to African Americans. It disagrees with what was reported by our study since the African population showed an AUC value of 0.81 compared to values of 0.78 or 0.73 for the Asian and Hispanic populations, respectively. Likewise, Baneu's [50] study reported cut-off points in the Asian population from 2.19 to 2.48 and in the Hispanic population from 2.80 to 3.20, unlike what was reported in our studies whose values were from 1.41 to 2.00 and from 2.00 to 2.22, respectively.

Despite this, both studies support Tg/HDL ratio is a simple and accessible marker for IR but recommends that more research is needed on personalized cut-off values that reflect ethnic, gender, and age group differences.

It is important to mention that the cut-off points must be interpreted correctly depending on the stage of puberty since when analyzing the TyG index and its association with insulin resistance it was found that, although the cut-off points are similar, the sensitivity and specificity were more precise in the prepubertal group (Formula A: S 92%, E 100%; Formula B: S 84%, E 100%), unlike adolescents (Formula A: S 72%, E 54%; Formula B: S 89%, E 69%) [36]. These findings would be explained by the role that growth hormone (GH) plays on insulin throughout children's development. There are various mechanisms by which GH ends up generating insulin resistance like inhibiting the phosphorylation of the insulin receptor, increasing the release of fatty acids through lipotoxicity that, by competing with glucose, prevents it from being captured by the muscles and stimulating gluconeogenesis, which stimulates greater secretion of glucose that saturates and, therefore, desensitizes insulin [51]. All these changes begin in childhood but reach their peak during puberty in a physiological and transitory manner since sexual maturation and the greater requirement for energy and muscle development stimulate the pituitary gland to secrete GH and the growth factor. insulin (IGF-1) [52].

**Table 5. GRADE summary of findings.**

| Index test | № of participants [studies] | Outcome | Effect size | GRADE certainty assessment | | | | | Certainty of the evidence |
|---|---|---|---|---|---|---|---|---|---|
| | | | | Risk of bias | Indirectness | Inconsistency | Imprecision | Publication bias | |
| *Insulin resistance in obesity or overweight children/adolescents* | | | | | | | | | |
| TyG | 2978 [5 observational studies] | Sensitivity | Range 60.0 to 92.0 | Very Serious[a] | Not serious | Serious[b] | Not serious | Not serious | ⊕○○○ Very Low |
| | 2978 [5 observational studies] | Specificity | Range 54.0 to 100.0 | Very Serious[a] | Not serious | Serious[b] | Not serious | Not serious | ⊕○○○ Very Low |
| Tg/HDL | 8416 [9 observational studies] | Sensitivity | Range 14.8 to 89.0 | Very Serious[a] | Not serious | Serious[b] | Not serious | Not serious | ⊕○○○ Very Low |
| | 8416 [9 observational studies] | Specificity | Range 45.0 to 97.6 | Very Serious[a] | Not serious | Serious[b] | Not serious | Not serious | ⊕○○○ Very Low |
| *Insulin resistance in General children/adolescents* | | | | | | | | | |
| TyG | 18064 [8 observational studies] | Sensitivity | Range 62.0 to 88.5 | Very Serious[a] | Not serious | Serious[b] | Not serious | Not serious | ⊕○○○ Very Low |
| | 18064 [8 observational studies] | Specificity | Range 50.5 to 81.0 | Very Serious[a] | Not serious | Serious[b] | Not serious | Not serious | ⊕○○○ Very Low |
| Tg/HDL | 718 [3 observational studies] | Sensitivity | Range 68.0 to 94.0 | Very Serious[a] | Not serious | Serious[b] | Not serious | Not serious | ⊕○○○ Very Low |
| | 718 [3 observational studies] | Specificity | Ranged 48.0 to 86.0 | Very Serious[a] | Not serious | Serious[b] | Not serious | Not serious | ⊕○○○ Very Low |

GRADE Working Group grades of evidence

High certainty: we are very confident that the true effect lies close to that of the estimate of the effect.

Moderate certainty: we are moderately confident in the effect estimate: the true effect is likely to be close to the estimate of the effect, but there is a possibility that it is substantially different.

Low certainty: our confidence in the effect estimate is limited: the true effect may be substantially different from the estimate of the effect.

Very low certainty: we have very little confidence in the effect estimate: the true effect is likely to be substantially different from the estimate of effect.

Explanations

[a] <50% of the of the studies had a low risk of bias.

[b] Values of studies are heterogeneous among themselves.

Therefore, like our results, it has been found that insulin resistance is higher in puberty compared to non-puberty and that this difference is exacerbated by up to 40% if we refer to children with a diagnosis of diabetes [53]. When studying both diabetic and non-diabetic pubertal children, an inverse relationship was observed between 24-hour levels of growth hormone and the insulin response (r = −0.52, p = 0.01), thus explaining the difficulty in the management of this group [54]. Furthermore, when a standard glucose level of 125 mg/dL was administered, the levels of C-peptide, which measures insulin secretion, were 2–3 times higher in pubertal children as opposed to prepubertal children, which expresses poor insulin sensitivity to normal glucose values [53]. In the pubertal subgroup, it was also found that insulin resistance predominates in Tanner stages 1 and 2, it returns to levels almost similar to the prepubertal subgroup in Tanner stage 5 and that, in all stages, resistance was higher in the female sex [55]. Although the hyperinsulinemic-euglycemic clamp is the gold standard for measuring insulin resistance in children, the TyG index is presented as a more practical and cheaper biomarker to apply even in health centers, so more studies are required to study this biomarker according to the stage of puberty, gender, ethnicity, and other determining factors such as body mass index and level of physical activity.

## Recommendations for future research

The certainty of the evidence of the TyG and Tg/HDL indices for diagnosing IR was very low mainly due to high risk of bias and heterogeneity. Therefore, we recommend conducting studies with higher methodological quality related to a

diagnostic accuracy study that meets the points related to the STARD reporting guidelines. In this study, 14 of the 21 included studies met the STARD guidelines. However, considering that this guide focuses on providing transparency and a better structure of the information necessary for subsequent analysis, it is possible that the results were not affected, at least in terms of statistical significance, since these depend on other factors such as the methodology and the sample size [56]. We also suggest conducting studies in European and Middle Eastern countries, since these regions were underrepresented in the present review even though we did not apply region restrictions in the search. Then, it could be because they have a higher level of insulin resistance despite controlling for factors such as body mass index or family history [57,58] and also because according to a survey carried out among doctors from European centers, they reported feeling more comfortable using traditional methods such as the oral glucose tolerance test and, at a biochemical level, they prefer to analyze insulin values, therefore that it is unlikely that there will be marked interest to study other biomarkers [59]. Finally, these indices could be incorporated into new or current diagnostic models to improve sensitivity and specificity in diseases derived from insulin resistance like cardiovascular diseases previously mentioned, or in the risk of developing metabolic syndrome since cut-off points have been found for the TyG index from 4.65 to 8.66 in people with a sedentary lifestyle but with questionable sensitivity and specificity values [60], so it is necessary to define confounders and establish more reference diagnostic tests.

### Clinical applicability

Our findings suggest that the TyG index could help identify a higher risk of developing metabolic diseases in both healthy children and adolescents and those with overweight or obesity. Other potential biomarkers of IR, such as adiponectin levels and HOMA-IR, have been proposed. However, these markers often require advanced equipment and preparation. In this context, the TyG index gains clinical importance as it is derived from simple, cost-effective tests that are routinely available. It can be used together with clinical or laboratory variables to generate valid predictive models.

The Tg/HDL index has proven to be a valuable tool in clinical practice for assessing cardiometabolic risk, particularly in patients with obesity or overweight. Several studies have explored its diagnostic accuracy in relation to insulin resistance, using HOMA-IR as a reference. Despite the heterogeneity in reported AUC values, which ranged from 0.687 to 0.809, its clinical utility as a marker of IR remains consistent. In general populations, Tg/HDL cutoff values between 1.41 and 2.22 have been reported, with sensitivities of up to 94% and specificities of up to 86%, further reinforcing its relevance in the early identification of metabolic dysfunctions. These findings suggest that the Tg/HDL index may be effective as a complementary method in the clinical assessment of patients at high risk of developing metabolic diseases, aiding in preventive therapeutic decision-making.

To facilitate the use of these indices in clinical and research settings, institutions can adopt standardized protocols that integrate the TyG and Tg/HDL indices into routine evaluations. For example, decision thresholds could be tailored to specific populations based on validated cut-off points derived from this and future research. Additionally, clinical workflows could incorporate automated calculations of these indices using electronic health record systems to ensure consistency and scalability.

### Limitations

This systematic review has some limitations. First, the high heterogeneity among studies prevented the performance of a meta-analysis, and thus, our results are not conclusive. In addition, we did not have access to regional databases such as those in China, but we conducted a systematic search without language or time restrictions. Furthermore, our search strategy did not find clinical trials that used both diagnostic tests, so the clinical impact could not be determined. However, because our objective was to determine the diagnostic accuracy of these tests, the selection of analytical studies was appropriate. Finally, most studies used the HOMA-IR as the gold standard instead of the HEC.

## Conclusions

Based on the very low certainty of evidence, we conclude that there is insufficient certainty to recommend the use of the TyG and Tg/HDL indices for the diagnosis of IR in children and adolescents. Studies with better methodological quality are needed, as well as research in European and middle eastern populations and their inclusion in diagnostic models.

## Supporting information

**S1. Table. PRISMA checklist.**
(DOCX)

**S2 Table. Search strategy.** Complete search strategy used for each database.
(DOCX)

**S3 Table. Decision table based on GRADE system.** Adaptation created by authors to assess certainty of evidence in narrative syntheses.
(DOCX)

**S4 Table. Studies evaluated in the selection process.** Includes all studies reviewed for inclusion, with reasons for exclusion when applicable.
(DOCX)

**S5 Table. Studies excluded.** Full list of excluded studies with justification.
(DOCX)

**S6 Table. Risk of bias (QUADAS-2) according to each domain.** Detailed results of QUADAS-2 assessments for each study included.
(DOCX)

## Acknowledgments

None

## Author contributions

**Conceptualization:** Miguel Cabanillas-Lazo, Carlos Quispe-Vicuña, Milagros Pascual-Guevara, Claudia Cruzalegui-Bazán, Arturo Duran-Pecho, José Paz-Ibarra, Victor Velásquez-Rimachi.

**Data curation:** Miguel Cabanillas-Lazo, Carlos Quispe-Vicuña, Milagros Pascual-Guevara, Claudia Cruzalegui-Bazán, José Paz-Ibarra, Victor Velásquez-Rimachi.

**Formal analysis:** Miguel Cabanillas-Lazo, Carlos Quispe-Vicuña, Milagros Pascual-Guevara, Arturo Duran-Pecho.

**Funding acquisition:** Victor Velásquez-Rimachi.

**Investigation:** Miguel Cabanillas-Lazo, Carlos Quispe-Vicuña, Milagros Pascual-Guevara, Claudia Cruzalegui-Bazán, Arturo Duran-Pecho, José Paz-Ibarra.

**Methodology:** Miguel Cabanillas-Lazo, Carlos Quispe-Vicuña, Milagros Pascual-Guevara, Claudia Cruzalegui-Bazán, Arturo Duran-Pecho, José Paz-Ibarra, Victor Velásquez-Rimachi.

**Project administration:** Victor Velásquez-Rimachi.

**Resources:** Victor Velásquez-Rimachi.

**Software:** Miguel Cabanillas-Lazo, Carlos Quispe-Vicuña.

**Writing – original draft:** Miguel Cabanillas-Lazo, Carlos Quispe-Vicuña, Milagros Pascual-Guevara, Claudia Cruzalegui-Bazán, Arturo Duran-Pecho, José Paz-Ibarra, Victor Velásquez-Rimachi.

**Writing – review & editing:** Miguel Cabanillas-Lazo, Carlos Quispe-Vicuña, Milagros Pascual-Guevara, Claudia Cruzalegui-Bazán, Arturo Duran-Pecho, José Paz-Ibarra, Victor Velásquez-Rimachi.

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
