## [Decision Letter · Decision Letter 0]

PONE-D-24-47941Diagnostic accuracy of triglyceride to glucose index and triglyceride/high-density lipoprotein index for insulin resistance among children and adolescents: A systematic reviewPLOS ONE

Dear Dr. Velásquez Rimachi,

Thank you for submitting your manuscript to PLOS ONE. After careful consideration, we feel that it has merit but does not fully meet PLOS ONE’s publication criteria as it currently stands. Therefore, we invite you to submit a revised version of the manuscript that addresses the points raised during the review process.

Please make peer-to-peer modifications to the reviewer's comments.

We look forward to receiving your revised manuscript.

Kind regards,

Qian Wu

Academic Editor

PLOS ONE

Journal Requirements:

2. We note that your Data Availability Statement is currently as follows: [All relevant data are within the manuscript and its Supporting Information files.] Please confirm at this time whether or not your submission contains all raw data required to replicate the results of your study. Authors must share the “minimal data set” for their submission. PLOS defines the minimal data set to consist of the data required to replicate all study findings reported in the article, as well as related metadata and methods (https://journals.plos.org/plosone/s/data-availability#loc-minimal-data-set-definition). For example, authors should submit the following data: - The values behind the means, standard deviations and other measures reported; - The values used to build graphs; - The points extracted from images for analysis. Authors do not need to submit their entire data set if only a portion of the data was used in the reported study. If your submission does not contain these data, please either upload them as Supporting Information files or deposit them to a stable, public repository and provide us with the relevant URLs, DOIs, or accession numbers. For a list of recommended repositories, please see https://journals.plos.org/plosone/s/recommended-repositories. If there are ethical or legal restrictions on sharing a de-identified data set, please explain them in detail (e.g., data contain potentially sensitive information, data are owned by a third-party organization, etc.) and who has imposed them (e.g., an ethics committee). Please also provide contact information for a data access committee, ethics committee, or other institutional body to which data requests may be sent. If data are owned by a third party, please indicate how others may request data access.

3. We note you have included a table to which you do not refer in the text of your manuscript. Please ensure that you refer to Tables 3 and 4 in your text; if accepted, production will need this reference to link the reader to the Table.

4. Please include captions for your Supporting Information files at the end of your manuscript, and update any in-text citations to match accordingly. Please see our Supporting Information guidelines for more information: http://journals.plos.org/plosone/s/supporting-information .

5. We notice that your supplementary [Supplementary Tables 1-3] are included in the manuscript file. Please remove them and upload them with the file type 'Supporting Information'. Please ensure that each Supporting Information file has a legend listed in the manuscript after the references list.

6. As required by our policy on Data Availability, please ensure your manuscript or supplementary information includes the following:

Reviewers' comments:

Reviewer's Responses to Questions

**Comments to the Author**

1. Is the manuscript technically sound, and do the data support the conclusions?

Reviewer #1: Yes

Reviewer #2: Yes

Reviewer #3: Partly

Reviewer #4: Yes

Reviewer #5: Yes

Reviewer #6: Yes

2. Has the statistical analysis been performed appropriately and rigorously? 

Reviewer #1: Yes

Reviewer #2: Yes

Reviewer #3: I Don't Know

Reviewer #4: I Don't Know

Reviewer #5: Yes

Reviewer #6: Yes

3. Have the authors made all data underlying the findings in their manuscript fully available?

Reviewer #1: Yes

Reviewer #2: Yes

Reviewer #3: Yes

Reviewer #4: Yes

Reviewer #5: Yes

Reviewer #6: Yes

4. Is the manuscript presented in an intelligible fashion and written in standard English?

Reviewer #1: Yes

Reviewer #2: Yes

Reviewer #3: Yes

Reviewer #4: Yes

Reviewer #5: No

Reviewer #6: Yes

5. Review Comments to the Author

Reviewer #1: Insulin resistance (IR) is a prevalent metabolic disorder associated with obesity, type 2 diabetes, and cardiovascular disease. While the hyperinsulinemic-euglycemic clamp is considered the gold standard for IR diagnosis, it remains costly and complex. Consequently, simpler and more affordable measures, like the triglyceride-to-glucose (TyG) index and the triglyceride/high-density lipoprotein (Tg/HDL) index, have been investigated as alternative diagnostic tools for IR. This review by Rimachi et al. offers a thorough evaluation of the diagnostic utility of TyG and Tg/HDL indices on children and adolescents, which is especially relevant given the rise of insulin resistance (IR) in youth. This paper is well-structured and holds clinical significance. Below are my comments.

1. Beyond HEC and HOMA-IR, indices like HOMA2 and QUICKI also serve as useful surrogate measures of insulin resistance. Additionally, several metrics evaluate insulin resistance through serum glucose or insulin response to a glucose challenge. Adding more information on the clinical or screening applicability of these indices, especially highlighting their specific advantages and limitations, would enhance the paper.

2. In future research directions, including a brief discussion on the importance of conducting longitudinal studies to assess whether these indices can predict metabolic outcomes later in life.

3. Avoid repetitive grammatical structures to improve readability. For example, lines 406–412 could be rephrased.

Reviewer #2: 1. In the abstract,

-Add a sentence stating the gap in current knowledge that this review aims to fill, such as "There is a need for simpler, cost-effective indices to diagnose insulin resistance in pediatric populations."

-Briefly mention the criteria for study selection and any major exclusion criteria.

-The results section provides a range of sensitivity, specificity, and AUC values but does not interpret what these ranges mean for the diagnostic utility of the indices. Add a brief interpretation of the results, such as "These results suggested that while the TyG index generally shows higher diagnostic accuracy compared to the Tg/HDL index".

-Add the conclusion by suggesting specific areas for future research.

2. In the Introduction,

-While the introduction mentions various mechanisms and the public health impact of IR, it lacks a focused discussion on why the Tg/HDL and TyG indices were specifically chosen for this review.

3. In Methodology,

-Explain why two different formulas for the TyG index are considered and the potential impact on the study results.

-Add details on how each of the four domains in QUADAS-2 was evaluated and how potential biases were addressed in the review process.

4. In Results,

-The differences in the reference tests used (HEC, HOMA-IR, and Matsuda index) are mentioned without further explanation of their significance.

-Discuss how this heterogeneity affects the reliability of the diagnostic indices.

5. In Discussion and Conclusion,

-Please provide more details on how other researchers or institutions can adopt or adapt the decision table.

-Some of the references to prior studies (e.g., Chen et al., Baneu et al.) could be integrated more cohesively to highlight the strengths and weaknesses of previous findings.

-A clearer synthesis of how your results align or differ from prior studies should be mentioned.

Reviewer #3: General comments

1. Include the reference number in the sentence, before the full stop or comma instead of after, for better readability of the text. For example, in the first sentence of the introduction:

Insulin resistance (IR) is a metabolic condition caused by a reduction in the ability of cells to absorb and utilize blood glucose. (1)

Change to:

Insulin resistance (IR) is a metabolic condition caused by a reduction in the ability of cells to absorb and utilize blood glucose (1).

2. Make sure the reference style agrees with the journal’s guidelines both in the reference list and in the text. In the reference list the DOI of the articles is missing and change brackets type of in-text citation from (1) to [1]. Using () brackets is confusing when citations and a list with numbers are mentioned in one sentence, as is the case in line 186-192 under the heading “Narrative synthesis”

Comments on the Introduction:

3. The sentence “However, for their calculation insulin and glucose levels are determined in fasting plasma, but due to the instability of insulin in the patient, the blood collected must be processed and frozen immediately” on line 108 - 110 needs a citation. This is one of the main arguments for alternative diagnostic tests that his paper is reviewing.

4. The layout of figure 1 can be improved to improve readability.

Comments on the Discussion: The Discussion needs major revisions. In particular the authors have to highlight the novelty of their manuscript compared to already published review articles.

5. The paragraph “Pathophysiological mechanisms” line 306-335 does not belong in the discussion as this is not a discussion of the results. It should be incorporated in the introduction or deleted.

6. The statement on line 339-440 “This review found the Tg/HDL index to be a novel indicator of IR and an atherogenic index of plasma;” is a strong statement considering the results of the review (“Tg/HDL index had a sensitivity of 14.8%-85.7%” line 286). Also, the review does not mention the “atherogenic index of plasma” earlier in the text and it should be explained that this index is composed of triglycerides and high-density lipoprotein cholesterol. Furthermore, the review developed a systematic review to evaluate the evidence regarding the diagnostic accuracy of TG/HDL in the diagnosis of insulin resistance in children and adolescents, it did not evaluate the diagnostic accuracy of the atherogenic index or cardiovascular events. The relation of the results from the review to the study by Chen et al. (line 341-347) requires therefore further elaboration or should be rephrased. It should also be explained if Tg/HDL index applies to the general population or to obese/overweight individuals.

7. In paragraph "Comparison with other systematic reviews" the authors refer to two other systematic review, 56 & 57. The authors summarize the findings in these published articles but do not discuss their own results compared to the results in these two reviews. The authors should also discuss why their review is different from the already published reviews and argue it is not “more of the same”. This is my main concern to recommend acceptance of this manuscript.

Reviewer #4: This manuscript is a meaningful attempt to elucidate the diagnostic value of triglyceride to glucose index(TyG) and triglyceride/high-density lipoprotein index(Tg/HDL) as a surrogate marker of HOMA-IR or insulin resistance itself.

However, the attempt was not successful because of high heterogeneity of included studies. Actually, the authors could not even perform meta-analysis.

However, these inconclusive and somehow limited results still indicate TyG and Tg/HDL to be a promising simple screening marker of metabolic syndrome.

Although the results were negative, it is worth to be published I think.

Thank you for submit interesting manuscript.

Reviewer #5: This is a novel approach to improving our ability as a global society to increase accessibility to testing, diagnosis, and treatment. I'm delighted to see this appraoch

Introduction.

Line 93: Do you mean "precise" here instead of "accurate"? I think the intention is that we don't know the exact ratio here, but if the data are truly inaccurate, then I would suggest removing "three times" and instead using "much more" or something similar.

Line 95: I'm not sure what is meant by "in turn" - I think you mean "Additionally" and suggest repeating "IR" so that your readers know whether you are referring to IR or diabetes.

Line 97: I think you mean "can" instead of "usually." I will stop commenting on the language for most of the study, but would suggest having additional review of your wording so that your message is clearer.

Line 99: It would be helpful to underscore why it is important to diagnose IR, given that there pharmaceutical options are available only for related disorders, and it's unlikely that IR will exist (or be tested for) in the absence of other signs and symptoms

I'm delighted to see your explicit discussion of bias and applicability. However, while you acknowledge the risk of bias, it is difficult for me as a reader to understand its implications. Given the impressive geographic variation of your study, and the know genetic and environmental impacts on obesity an metabolic syndrome, I think this is an area that warrants further exploration if your goal is to summarize with a single recommendation (insufficient evidence). Could it be that for some populations this is not the case? Perhaps a more applicable study would be to identify which studies are relevant to whom? I don't know the answer here, but I think it is worth considering.

If you were to separate your data in subpopulations, you could perform stratified meta-analyses - and consider including AUC figure for some of the subpopulations would also be helpful. It also might be helpful to consider whether these indices have additional use - perhaps as a screening or as a risk factor in and of themselves. While your overall conclusions appears valid based on your methodology, given the extensive work and apparent need for increased access to diagnostic tools, it does seem that there are other implications to consider.

Overall, this is a compelling review and a very useful respository of a wide variety of studies. I think the authors add value to the state of the literature, however, I also believe that additional analysis could better direct future research as to whether these indices are likely to be useful for some in clinical practice, or whether despite years of study these laboratory values are simply not going to be precise enough to be useful.

Reviewer #6: Cost-effective tools for the diagnosis of insulin resistance in children are currently not yet well established. Two potential tools include measurement of triglyceride (Tg), high density lipoprotein (HDL) and glucose to measure TyG and Tg/HDL indices, which have been investigated across thousands of studies over the last 20 years in countries across the globe. The authors implemented a careful and robust selection process across multiple online databases, to analyse data from published studies investigating the TyG and/or Tg/HDL indices in children and adolescents, compared to recognised tools such as HEC or HOMA-IR. The authors have summarised the results, and calculated various measures (QUADAS-2, GRADE) to quantify certainty in the final results. I believe the review is worthy of publication after some minor revision. Please find below my point-by-point suggestions.

1) Methods: Eligibility criteria could be slightly clearer – do the studies need to meet any, or all, of the 3 criteria? In criterion 3, do the studies need to have evaluated diagnostic accuracy of both TyG and Tg/HDL indices, or at least one?

2) Results, Narrative synthesis: Tables 3 and 4 should be referenced here. The PPV and NPV values are never mentioned in the Results, Discussion or Conclusion, can they be removed altogether? If the authors believe the information to be important, it should be at least referenced in the text.

3) Discussion, Summary of the main results: The authors mention this systematic review as being with meta-analysis, despite earlier clarifying that they have performed a narrative synthesis.

4) Discussion, Pathophysiological mechanisms:

a. This section does not directly relate back to the findings of their systematic review, meaning it reads more like an introduction to the mechanisms. I think it would benefit from being partly moved to the Introduction, and then be referred back to within the discussion, in light of the results of this review.

b. The authors’ discussion about puberty is interesting. Perhaps it suggests a need to consider the data from pre-pubertal and adolescent children separately? I note from Table 3 that the Sanchez 2020 study in obese or overweight children analysed TyG for these two groups separately, and observe differences between sensitivity, specificity and AUC, albeit with similar TyG cutoffs – can the authors comment on this?

5) Discussion, Recommendations for future research:

a. The authors correctly recommend that future studies should be conducted that conform to points within the STARD reporting guidelines. How many of the studies included in this review were deemed to conform to the STARD reporting guidelines? Can any results be derived from only analysing those studies?

b. Readers may be surprised that European countries were underrepresented in this review – can the authors comment on why that is? Were European studies more commonly excluded during selection, if so why?

6) Discussion, Limitations: The authors suggest that the clinical impact of the use of both diagnostic tests could not be determined as they did not include clinical trials that used both tests. Can the authors explain why they did not include these trials, knowing that they could not then determine the clinical impact of both tests?

7) Discussion, Comparison with other systematic reviews: The authors state that “this review found the Tg/HDL index to be a novel indicator of IR”. However the authors later state in Conclusions that they “conclude that there is insufficient certainty to recommend the use of the … Tg/HDL ind[ex] for the diagnosis of IR in children and adolescents”. The earlier comment should be re-worded to avoid overstating their results.

8) Figure 1:

a. The numbers of records do not add up within the ‘Screening’ panel, e.g. of the 1135 records screened, 1077 were excluded, resulting in 57 records being sought for retrieval. However 1135 – 1077 = 58. If the numbers are indeed correct, the flow of the diagram needs to be made clearer.

b. Figure 1 needs tidying up in general. Arrows appear missing. Text is truncated in some text boxes. Text is poorly aligned within other text boxes.

9) Tables 1, 2 and 3: It would be helpful to readers if the authors could provide the reference ID in the tables, so that the reader does not need to look through the full references list to find the related study.

6. PLOS authors have the option to publish the peer review history of their article (what does this mean? ). If published, this will include your full peer review and any attached files.

**Do you want your identity to be public for this peer review?** For information about this choice, including consent withdrawal, please see our Privacy Policy .

Reviewer #1: No

Reviewer #2: No

Reviewer #3: No

Reviewer #4: No

Reviewer #5: No

Reviewer #6: **Yes: ** Christopher Smith

---

## [Author Response · Author response to Decision Letter 1]

21 Apr 2025

POINT-BY-POINT RESPONSE TO REVIEWERS

Manuscript: PONE-D-24-47941 “Diagnostic accuracy of triglyceride to glucose index and triglyceride/high-density lipoprotein index for insulin resistance among children and adolescents: A systematic review”

Dear Qian Wu

Academic Editor

PLOS ONE

We would like to thank you and the reviewers for your helpful comments and suggestions, which we agree helped us to improve the manuscript.

We include here a point-by-point response to all comments after taking these observations into account. In addition, we have uploaded on the website:

1) A file containing the reviewers’ suggestions and our specific responses to their comments.

2) The revised manuscript with the changes marked using highlights in Microsoft Word.

We hope the article is now suitable for publication in PLOS ONE and we look forward to hearing from you.

Yours sincerely,

Victor Velasquez-Rimachi MD, MSc.

Corresponding Author

RESPONSE TO REVIEWERS

Reviewer #1: Insulin resistance (IR) is a prevalent metabolic disorder associated with obesity, type 2 diabetes, and cardiovascular disease. While the hyperinsulinemic-euglycemic clamp is considered the gold standard for IR diagnosis, it remains costly and complex. Consequently, simpler and more affordable measures, like the triglyceride-to-glucose (TyG) index and the triglyceride/high-density lipoprotein (Tg/HDL) index, have been investigated as alternative diagnostic tools for IR. This review by Rimachi et al. offers a thorough evaluation of the diagnostic utility of TyG and Tg/HDL indices on children and adolescents, which is especially relevant given the rise of insulin resistance (IR) in youth. This paper is well-structured and holds clinical significance. Below are my comments.

1. Beyond HEC and HOMA-IR, indices like HOMA2 and QUICKI also serve as useful surrogate measures of insulin resistance. Additionally, several metrics evaluate insulin resistance through serum glucose or insulin response to a glucose challenge. Adding more information on the clinical or screening applicability of these indices, especially highlighting their specific advantages and limitations, would enhance the paper.

Response: Thank you for your valuable suggestion. We acknowledge that indices like HOMA2 and QUICKI, along with other insulin resistance metrics based on glucose challenge responses, are relevant surrogate measures. To address this, we have included a brief mention of these indices in the introduction, highlighting their applicability, advantages, and limitations.

2. In future research directions, including a brief discussion on the importance of conducting longitudinal studies to assess whether these indices can predict metabolic outcomes later in life.

Response: Dear reviewer, we have made the corresponding changes.

3. Avoid repetitive grammatical structures to improve readability. For example, lines 406–412 could be rephrased.

Response: Dear reviewer, we have made the corresponding changes.

Reviewer #2: 1. In the abstract,

-Add a sentence stating the gap in current knowledge that this review aims to fill, such as "There is a need for simpler, cost-effective indices to diagnose insulin resistance in pediatric populations."

-Briefly mention the criteria for study selection and any major exclusion criteria.

-The results section provides a range of sensitivity, specificity, and AUC values but does not interpret what these ranges mean for the diagnostic utility of the indices. Add a brief interpretation of the results, such as "These results suggested that while the TyG index generally shows higher diagnostic accuracy compared to the Tg/HDL index".

-Add the conclusion by suggesting specific areas for future research.

Response: Dear reviewer, we have made the corresponding changes in the abstract.

2. In the Introduction,

-While the introduction mentions various mechanisms and the public health impact of IR, it lacks a focused discussion on why the Tg/HDL and TyG indices were specifically chosen for this review.

Response: Dear reviewer, we have made the corresponding changes in the introduction.

Line 91-93: “For these reasons, the use of the Tg/HDL index and Tg-Glucose (TyG) index as cost-effective predictors or markers of IR in children and adolescents has been reported in different regions”

3. In Methodology,

-Explain why two different formulas for the TyG index are considered and the potential impact on the study results.

Response: Dear reviewer, we have made the corresponding changes in the methodology.

Line 111-116: “The following formulas for TyG were considered:

Formula A: ln [Tg (mg/dL) × fasting glucose (mg/dL) /2]

Formula B: ln [Tg (mg/dL) × fasting glucose (mg/dL)] /2

The following formula for Tg/HDL was considered:

Formula C: Tg(mg/dL) / HDL (mg/d)”

-Add details on how each of the four domains in QUADAS-2 was evaluated and how potential biases were addressed in the review process.

Response: Dear reviewer, we have made the corresponding changes in the methodology.

Line 152-157: “Bias in patient selection was addressed by excluding studies with non-representative populations. For the index test, we ensured that TyG and Tg/HDL indices were measured consistently with clear definitions. Reference standard bias was minimized by including studies that used validated methods such as HOMA-IR or HEC. Finally, flow and timing issues were mitigated by including only studies with consistent protocols for index and reference testing. Any disagreements in bias assessment were resolved by consensus.”

4. In Results,

-The differences in the reference tests used (HEC, HOMA-IR, and Matsuda index) are mentioned without further explanation of their significance.

Response: Thanks for the commentary, we added some specifications in the section Characteristics of studies.

-Discuss how this heterogeneity affects the reliability of the diagnostic indices.

Response: Thanks for the commentary, we provided more details in discussion.

5. In Discussion and Conclusion,

-Please provide more details on how other researchers or institutions can adopt or adapt the decision table.

Response: Thanks for the commentary, we provided more details in discussion.

-Some of the references to prior studies (e.g., Chen et al., Baneu et al.) could be integrated more cohesively to highlight the strengths and weaknesses of previous findings.

Response: Dear reviewer, we have made the corresponding changes in the references.

-A clearer synthesis of how your results align or differ from prior studies should be mentioned

Response: Dear reviewer, we have made new redaction to clarify the comparison with preview studies. 

Reviewer #3: General comments

1. Include the reference number in the sentence, before the full stop or comma instead of after, for better readability of the text. For example, in the first sentence of the introduction: Insulin resistance (IR) is a metabolic condition caused by a reduction in the ability of cells to absorb and utilize blood glucose. (1)

Change to: Insulin resistance (IR) is a metabolic condition caused by a reduction in the ability of cells to absorb and utilize blood glucose (1).

Response: Dear reviewer, we have accepted the commentary and made the corresponding changes.

2. Make sure the reference style agrees with the journal’s guidelines both in the reference list and in the text. In the reference list the DOI of the articles is missing and change brackets type of in-text citation from (1) to [1]. Using () brackets is confusing when citations and a list with numbers are mentioned in one sentence, as is the case in line 186-192 under the heading “Narrative synthesis”

Response: Dear reviewer, we have accepted the commentary and made the corresponding changes.

Comments on the Introduction:

3. The sentence “However, for their calculation insulin and glucose levels are determined in fasting plasma, but due to the instability of insulin in the patient, the blood collected must be processed and frozen immediately” on line 108 - 110 needs a citation. This is one of the main arguments for alternative diagnostic tests that his paper is reviewing.

Response: Dear reviewer, we added the corresponding reference.

4. The layout of figure 1 can be improved to improve readability.

Response: Dear reviewer, we have accepted the commentary and clarify figure 1 content.

Comments on the Discussion: The Discussion needs major revisions. In particular the authors have to highlight the novelty of their manuscript compared to already published review articles.

Response: Dear reviewer, we have accepted the commentary and made the corresponding changes in section Comparison with other systematic reviews (Line 279).

5. The paragraph “Pathophysiological mechanisms” line 306-335 does not belong in the discussion as this is not a discussion of the results. It should be incorporated in the introduction or deleted.

Response: Dear reviewer, we have accepted the commentary and deleted the section Pathophysiological mechanisms.

6. The statement on line 339-440 “This review found the Tg/HDL index to be a novel indicator of IR and an atherogenic index of plasma;” is a strong statement considering the results of the review (“Tg/HDL index had a sensitivity of 14.8%-85.7%” line 286). Also, the review does not mention the “atherogenic index of plasma” earlier in the text and it should be explained that this index is composed of triglycerides and high-density lipoprotein cholesterol. Furthermore, the review developed a systematic review to evaluate the evidence regarding the diagnostic accuracy of TG/HDL in the diagnosis of insulin resistance in children and adolescents, it did not evaluate the diagnostic accuracy of the atherogenic index or cardiovascular events. The relation of the results from the review to the study by Chen et al. (line 341-347) requires therefore further elaboration or should be rephrased. It should also be explained if Tg/HDL index applies to the general population or to obese/overweight individuals.

Response: Dear reviewer, we have accepted the commentary and made the corresponding changes.

7. In paragraph "Comparison with other systematic reviews" the authors refer to two other systematic review, 56 & 57. The authors summarize the findings in these published articles but do not discuss their own results compared to the results in these two reviews. The authors should also discuss why their review is different from the already published reviews and argue it is not “more of the same”. This is my main concern to recommend acceptance of this manuscript.

Response: Dear reviewer, we have accepted the commentary and made the corresponding changes in section Comparison with other systematic reviews.

Reviewer #4: This manuscript is a meaningful attempt to elucidate the diagnostic value of triglyceride to glucose index(TyG) and triglyceride/high-density lipoprotein index(Tg/HDL) as a surrogate marker of HOMA-IR or insulin resistance itself.

However, the attempt was not successful because of high heterogeneity of included studies. Actually, the authors could not even perform meta-analysis.

However, these inconclusive and somehow limited results still indicate TyG and Tg/HDL to be a promising simple screening marker of metabolic syndrome.

Although the results were negative, it is worth to be published I think.

Thank you for submit interesting manuscript.

Response: Dear reviewer, we appreciate the commentary.

Reviewer #5: This is a novel approach to improving our ability as a global society to increase accessibility to testing, diagnosis, and treatment. I'm delighted to see this approach

Introduction.

Line 93: Do you mean "precise" here instead of "accurate"? I think the intention is that we don't know the exact ratio here, but if the data are truly inaccurate, then I would suggest removing "three times" and instead using "much more" or something similar.

Response: Dear reviewer, we have made the corresponding changes.

Line 95: I'm not sure what is meant by "in turn" - I think you mean "Additionally" and suggest repeating "IR" so that your readers know whether you are referring to IR or diabetes.

Response: Dear reviewer, we have made the corresponding changes.

Line 97: I think you mean "can" instead of "usually." I will stop commenting on the language for most of the study, but would suggest having additional review of your wording so that your message is clearer.

Response: Dear reviewer, we have made the corresponding changes.

Line 99: It would be helpful to underscore why it is important to diagnose IR, given that there pharmaceutical options are available only for related disorders, and it's unlikely that IR will exist (or be tested for) in the absence of other signs and symptoms

Response: Dear reviewer, we have made the corresponding changes.

I'm delighted to see your explicit discussion of bias and applicability. However, while you acknowledge the risk of bias, it is difficult for me as a reader to understand its implications. Given the impressive geographic variation of your study, and the know genetic and environmental impacts on obesity an metabolic syndrome, I think this is an area that warrants further exploration if your goal is to summarize with a single recommendation (insufficient evidence). Could it be that for some populations this is not the case? Perhaps a more applicable study would be to identify which studies are relevant to whom? I don't know the answer here, but I think it is worth considering.

Response: Dear reviewer, we have made the corresponding changes to clarify.

If you were to separate your data in subpopulations, you could perform stratified meta-analyses - and consider including AUC figure for some of the subpopulations would also be helpful. It also might be helpful to consider whether these indices have additional use - perhaps as a screening or as a risk factor in and of themselves. While your overall conclusions appears valid based on your methodology, given the extensive work and apparent need for increased access to diagnostic tools, it does seem that there are other implications to consider.

Overall, this is a compelling review and a very useful respository of a wide variety of studies. I think the authors add value to the state of the literature, however, I also believe that additional analysis could better direct future research as to whether these indices are likely to be useful for some in clinical practice, or whether despite years of study these laboratory values are simply not going to be precise enough to be useful.

Response: Dear reviewer, we appreciate your comment, but we did not have the possibility to perform a meta-analysis in this article. For future studies we will consider your important suggestion.

Reviewer #6: Cost-effective tools for the diagnosis of insulin resistance in children are currently not yet well established. Two potential tools include measurement of triglyceride (Tg), high density lipoprotein (HDL) and glucose to measure TyG and Tg/HDL indices, which have been investigated across thousands of studies over the last 20 years in countries across the globe. The authors implemented a careful and robust selection process across multiple online databases, to analyse data from published studies investigating the TyG and/or Tg/HDL indices in children and adolescents, compared to recognised tools such as HEC or HOMA-IR. The authors have summarised the results, and calculated various measures (QUADAS-2, GRADE) to quantify certainty in the final results. I believe the review is worthy of publication after some minor revision. Please find below my point-by-point suggestions.

1) Methods: Eligibility criteria could be slightly clearer – do the studies need to meet any, or all, of the 3 criteria? In criterion 3, do the studies need to have evaluated diagnostic accuracy of both TyG and Tg/HDL indices, or at least one?

Response: Dear reviewer, we have accepted the commentary and made the corresponding changes to clarify.

2) Results, Narrative synthesis: Tables 3 and 4 should be referenced here. The PPV and NPV values are never mentioned in the Results, Discussion or Conclusion, can they be removed altogether? If the authors believe the information to be import

---

## [Decision Letter · Decision Letter 1]

Diagnostic accuracy of triglyceride to glucose index and triglyceride/high-density lipoprotein index for insulin resistance among children and adolescents: A systematic review

PONE-D-24-47941R1

Dear Dr. Velásquez Rimachi,

We’re pleased to inform you that your manuscript has been judged scientifically suitable for publication and will be formally accepted for publication once it meets all outstanding technical requirements.

Kind regards,

Qian Wu

Academic Editor

PLOS ONE

Additional Editor Comments (optional):

Reviewers' comments:

Reviewer's Responses to Questions

**Comments to the Author**

1. If the authors have adequately addressed your comments raised in a previous round of review and you feel that this manuscript is now acceptable for publication, you may indicate that here to bypass the “Comments to the Author” section, enter your conflict of interest statement in the “Confidential to Editor” section, and submit your "Accept" recommendation.

Reviewer #1: All comments have been addressed

Reviewer #6: All comments have been addressed

2. Is the manuscript technically sound, and do the data support the conclusions?

Reviewer #1: Yes

Reviewer #6: (No Response)

3. Has the statistical analysis been performed appropriately and rigorously? 

Reviewer #1: Yes

Reviewer #6: (No Response)

4. Have the authors made all data underlying the findings in their manuscript fully available?

Reviewer #1: Yes

Reviewer #6: (No Response)

5. Is the manuscript presented in an intelligible fashion and written in standard English?

Reviewer #1: Yes

Reviewer #6: (No Response)

6. Review Comments to the Author

Reviewer #1: (No Response)

Reviewer #6: (No Response)

7. PLOS authors have the option to publish the peer review history of their article (what does this mean? ). If published, this will include your full peer review and any attached files.

**Do you want your identity to be public for this peer review?** For information about this choice, including consent withdrawal, please see our Privacy Policy .

Reviewer #1: No

Reviewer #6: **Yes: ** Christopher Smith

---

## [Editor Report · Acceptance letter]

PONE-D-24-47941R1

PLOS ONE

Dear Dr. Velásquez-Rimachi,

I'm pleased to inform you that your manuscript has been deemed suitable for publication in PLOS ONE. Congratulations! Your manuscript is now being handed over to our production team.

Kind regards,

on behalf of

Dr. Qian Wu

Academic Editor

PLOS ONE